# Can Wearable Cameras Be Used to Validate School-Aged Children’s Lifestyle Behaviours?

**DOI:** 10.3390/children6020020

**Published:** 2019-02-01

**Authors:** Bethan Everson, Kelly A. Mackintosh, Melitta A. McNarry, Charlotte Todd, Gareth Stratton

**Affiliations:** 1MRC Epidemiology Unit, University of Cambridge, School of Clinical Medicine, Institute of Metabolic Science, Cambridge Biomedical Campus, Cambridge CB2 0QQ, UK; Bethan.Everson@mrc-epid.cam.ac.uk; 2Applied Sports Technology Exercise and Medicine (A-STEM) Research Centre, College of Engineering, Swansea University, Bay Campus, Fabian Way, Swansea SA1 8EN, UK; k.mackintosh@swansea.ac.uk (K.A.M.); m.mcnarry@swansea.ac.uk (M.A.M.); 3College of Medicine, Data Science Building, Swansea University, Singleton Park, Swansea SA28PP, UK; c.e.todd@swansea.ac.uk

**Keywords:** wearable cameras, autographer, child’s health and activity tool (CHAT), health and lifestyle behaviours, parent-child dyad, observation, self-report, previous day recall

## Abstract

Wearable cameras combined with accelerometers have been used to estimate the accuracy of children’s self-report of physical activity, health-related behaviours, and the contexts in which they occur. There were two aims to this study; the first was to validate questions regarding self-reported health and lifestyle behaviours in 9–11-year-old children using the child’s health and activity tool (CHAT), an accelerometer and a wearable camera. Second, the study sought to evaluate ethical challenges associated with taking regular photographs using a wearable camera through interviews with children and their families. Fourteen children wore an autographer and hip-worn triaxial accelerometer for the waking hours of one school and one weekend day. For both of these days, children self-reported their behaviours chronologically and sequentially using the CHAT. Data were examined using limits of agreement and percentage agreement to verify if reference methods aligned with self-reported behaviours. Six parent–child dyads participated in interviews. Seven, five, and nine items demonstrated good, acceptable, and poor validity, respectively. This demonstrates that the accuracy of children’s recall varies according to the behaviour or item being measured. This is the first study to trial the use of wearable cameras in assessing the concurrent validity of children’s physical activity and behaviour recall, as almost all other studies have used parent proxy reports alongside accelerometers. Wearable cameras carry some ethical and technical challenges, which were examined in this study. Parents and children reported that the autographer was burdensome and in a few cases invaded privacy. This study demonstrates the importance of adhering to an ethical framework.

## 1. Introduction

In recent years, there has been limited success in reversing or curbing rates of obesity [1]. Efforts have predominantly focused on the energy balance equation, postulating that obesity is a result of positive energy imbalance [2] and therefore concentrating on independently examining physical activity [3] or dietary behaviours [4]. Whilst advances in technology have significantly improved the understanding of physical activity intensities and their distribution throughout the day [5], the accurate assessment of health behaviours in large samples of younger children poses many challenges. Commonly used measures of health behaviours such as paper-based questionnaires are prone to bias, relying on children’s cognitive ability to recall physical activity (type, mode, duration) [6] and dietary behaviours (type and amount of foods) [7]. Consequently, when collecting data on large populations, self-report web-based questionnaires that are highly engaging, low-cost, and minimise respondent and data input burden are commonly used with younger children [8,9,10,11,12]. The child’s health and activity tool (CHAT) is an example of a web-based questionnaire developed and designed for primary school children to produce a quick and easy method of gathering data on an array of health behaviours in 9–11-year-olds [13]. Items include wellbeing, sleep, nutrition, and physical activity. The CHAT can be completed on a computer, laptop or iPad and is administered in the primary school setting in the presence of researchers. Web-based tools offer a new and exciting measurement, as they are highly engaging for younger age groups, low-cost, and minimise respondent and data input burden [8,9,14]. Furthermore, an interface can be developed which is appealing to children, with visual aids and pictures included to improve recall [15]. To date, neither the CHAT nor any other children’s lifestyle questionnaires have been validated against objective measures, such as wearable cameras and accelerometers.

In addition to web-based questionnaires, wearable cameras have been used to capture behaviours and the contexts in which they occur [16,17]. The use of such devices has been advocated to document children’s food purchasing habits and exposure to food marketing [18,19]. Moreover, physical activity behaviours have been measured, with wearable cameras being used to examine transport behaviours in adults [20,21] and journey pursuits in school children [22,23]. More recently, cameras have been used to assess children’s exposure to green spaces, transport, and smoking [24]. The advantage of wearable technologies, such as cameras, is that daily behaviours can be examined through a first-person perspective [25,26]. An autographer is an example of this. It is a commercially available camera worn on a lanyard [26]. Quantifying time spent on certain activities and where they took place is important when investigating health-related behaviours. An overriding strength of objectively measured behaviours is overcoming the associated recall bias with self-reports [27].

The limitation with self-report tools is that they have used parent report as a proxy to validate children’s health-related behaviours [28,29,30,31,32]. To improve validation methods, wearable cameras offer a feasible technique from which comparisons can be made with web-based questionnaires, although ethical issues about their use in public and family settings exist. This paper sets out to address these issues. The primary aim of this study was to validate self-reported health and lifestyle behaviours in 9–11-year-old children using the CHAT with objective data captured using accelerometry and wearable cameras. Furthermore, the secondary aim was to identify the ethical challenges of using an autographer in children’s health-related research.

## 2. Materials and Methods

### 2.1. Participants and Settings

From January through to March 2015, four secondary and eight primary schools in South Wales were contacted via a telephone call and a follow-up email and invited to participate in the study. These schools were geographically located near the research institute. Of the twelve schools contacted, nine did not respond (six primary and three secondary schools), two declined (one primary and one secondary school), and one primary school agreed to participate. The participating school was a middle-class state school, consisting of approximately 350 pupils. A video outlining the study was shown to Year Five and Six pupils to enhance recruitment. Information sheets were subsequently disseminated. After 2 weeks, an information evening was held for parents and children to address concerns over the autographer via a question and answer session. Comments were recorded and transcribed verbatim for qualitative analysis. As a result, from the 83 pupils in years five and six, 14 (9 boys, 5 girls; 6 year five, 8 year six pupils; aged 11.0 ± 0.4 years volunteered to take part (17% response rate), providing written informed assent and parental consent. Headteacher consent was also provided on behalf of the school. One participant reported having a speech and language disability; the remaining participants reported having no physical or learning impairments.

### 2.2. Description and Validation of the CHAT

The CHAT includes 29 items examining the health and lifestyle behaviours of children aged 9 to 11 years old. Children are required to recall their previous day’s physical activity and sedentary behaviour, through a timeline of the respondent’s typical school day. It involves the following time segments: Before lessons, during and after school, which comprises items 1 to 14. Items were collapsed into standalone items if they contained several questions. This resulted in 21 items being examined in the present study. The remaining 15 items were not validated, including the seven-day recall of physical activity, dietary and lifestyle activities, habitual behaviours, movement skills, and perceptions of health, fitness, and wellbeing. The reason only 21 of the items were validated was because the latter section of the questionnaire related to child happiness with health, fitness, friends, family, and school, as well as general autonomy and competence. Additionally, part of the questionnaire referred to a weekly recall of health and lifestyle behaviours. Assessing weekly behaviour would have been very burdensome on children of this age. Consequently, to make the current study feasible, only items pertaining to previous day recall and health questions were validated. The concurrent validity of 21 items (Table 1) was tested using a hip-worn accelerometer and an autographer as reference methods.

### 2.3. Instrumentation and Procedures

All participants were familiarised with the autographer, hip-worn accelerometer, and study procedures and asked to wear both for waking hours of one school and one weekend day. Children were required to complete the CHAT the day after wearing the monitors. The autographer is a hands-free camera worn around the neck on an adjustable lanyard (3.7 × 2.3 × 9 cm, 59 g, Oxford Metrics Group), which automatically captures time-stamped first-person point-of-view images without any audio sound. Images are captured when a change is experienced in one of the five in-built sensors: A magnetometer, colour sensor, accelerometer, temperature or global positioning system. The autographer was set using a medium capture rate and disabled so participants could not alter them. This equated to pictures being taken every 15 s. If sensors were activated, then it would take a photograph more frequently. This was chosen as it balanced capture rate with image quality and battery life. To objectively measure physical activity, a hip-worn ActiGraph GT3X+ (ActiGraph; Pensacola, US, 10 g, 4.6 × 3.3 × 1.5 cm) was worn on an elastic belt at the right midaxillary line, at the level of the iliac crest [33]. Data were processed in 5 s epochs [34,35] using Evenson cut-points [36] and wear-time validation parameters according to Choi et al. [37]. All wear-time was cross-checked with participant diaries that provided removal and placement times. An eight-hour period was chosen for wear-time, as this represented the average battery life during the pilot study obtained using the medium capture rate. This allowed for six hours of the school day to be gathered, plus one hour before and after school.

The researcher met the participants at the school to collect both monitors the day after the observation. Time-stamped images from the autographer were downloaded into a custom-made browser [38]. Only school-day observations were included in the analysis. Fourteen children were recruited in the study. Of these, nine met the eight-hour wear-time criteria following initial data collection. However, two (of the five) initially not meeting the wear-time criteria agreed to rewear the monitors and subsequently met the criteria. Both participants completed the CHAT again the following day. Due to challenges with compliance and logistics with the autographer, analysis was conducted on an item-by-item basis. If either device was not worn for the given time frame relevant to the question, then the data were excluded from the analyses. Sample sizes for each item are shown in Table 1 and Table 2. For instance, due to a poor battery life, only six children had complete data sets for the item regarding sleep time and could be included in the analysis. This demonstrates that although 14 children were recruited on the study, only seven CHAT items used all 14 data sets in the analysis. This was to ensure the data collected were valid and therefore reflected the certainty that participants could accurately recall a given behaviour.

Once all photographs were downloaded, at the earliest convenience, each participant and one parent were invited to attend an image screening. The images were viewed in private and participants were instructed to delete any images they were not comfortable with, adhering to an ethical framework [39]. The time-stamp and the reason the image was deleted was recorded. Parents subsequently signed consent to confirm the remaining images could be analysed. During this phase, feedback was sought regarding their study experience and transcribed verbatim. After data were analysed, feedback sheets summarising results were provided to participants and a recruitment presentation inviting them to take part in follow-up interviews.

### 2.4. Method Comparison Protocol

Two annotation protocols were used to analyse physical activity and dietary behaviours. Protocols were adopted from previously validated direct observation protocols to ensure that missing and ambiguous data were handled consistently. Adapted from the System for Observing Children’s Physical Activity and Relationships during Play (SOCARP) protocol by Ridgers et al. [40], physical activity was classified as; sedentary (lying, sitting or standing), walking, moderate, and uncodeable. The second reference method was examined to ensure the self-reported physical activity intensity aligned with the most time spent at a given intensity (sedentary, light or moderate to vigorous physical activity (MVPA). For verification of mode of transport, the autographer images inferred the timing and context of the journey. This information was then used to examine the accelerometry trace to infer the journey mode.

Dietary behaviours were categorised as food preparation, eating a meal, snacking, and uncodeable. This was adapted from Chen et al.’s protocol [41]. An eating behaviour occurrence was defined as the presence of food or drink being observed. Food and drink types were recorded, and omissions were recorded if behaviours were falsely recorded. For fruit and vegetable servings, individual items were recorded and aggregated to achieve a total daily serving.

Screen-based activities were divided into four categories: TV, video games, iPad, and computer use. Duration of total screen activities was aggregated. Applying the annotations, images were segmented into events aligning with other methodologies [19,21]. Indiscernible images were annotated as uncodeable. Studies lack consensus as to what constitutes the start and end of an event [18,42]. For this study, the start of an event was defined as when the first image identified the start of behaviour. The end was defined as when an image displays a new behaviour occurring but not the final image of the previous behaviour.

A criterion for each item was designed to evaluate the extent that the self-report matched the reference method. To validate dietary behaviours, observed meals and drinks with the autographer were compared to the self-report. Three categories were used to determine the level of agreement; matches, reflecting self-reported foods/drinks that were observed; intrusions, reflecting self-reported foods/drinks that were not observed, which included misreporting the food or drink type; and omissions, which were when observed foods/drinks consumed were not reported. Drink type could not always be identified, but the presence of a drink was recorded and if identified, the drink was classified. For fruit and vegetable items, a count was defined as when a serving was photographed. Agreement was met if the self-report was within 1 count of that observed.

For physical activity items, agreement was met if children reported a duration that was within ±10 min (min) of the time observed in MVPA with both the autographer and accelerometer. Furthermore, agreement was confirmed if the intensities derived from the accelerometer and autographer aligned with the self-report. For this population, intensity was defined with time spent in light physical activity, MVPA or sedentary behaviour inferred by the accelerometer, which was cross-checked with the time-stamp on the autographer. Therefore, visual information was classed as sedentary (lying, sitting, standing), walking or moderate, and confirmed by the accelerometer. Consequently, researchers could confidently allocate an intensity of physical activity using both reference methods.

For screen use and homework, agreement was met if the duration estimated was within ±10 and 15 min, respectively. These boundaries were chosen as exploratory studies have shown children recall in blocks of 15 min or less [43,44]. For mode and context of the journey to school, agreement was met if the reported journey was photographed and confirmed by the accelerometry trace. For oral hygiene, agreement was met if participants estimated within 1 count of their observed or log book entries. For sleep patterns, agreement was met if the self-report was within ±30 min. This was to allow for privacy, which concurs with other studies [45].

### 2.5. Instrumentation of Group Interviews

A total of six parent–child dyad group interviews were conducted using an iterative questioning style. Predetermined topics included the following: Motivations and experiences of taking part, experiences wearing and observing the autographer, and overall perceptions of the autographer. On average, interviews lasted 23 min (±6 min). To facilitate discussion, a novel approach was used whereby parent–child dyads viewed images gathered to prompt memory and facilitate reflection. All group interviews were recorded using a digital recorder and transcribed verbatim. In addition, commentary obtained during the recruitment and image screening process was transcribed and combined with the group interview data.

### 2.6. Data Analysis

At the group level, each individual item from CHAT was cross-referenced with data from the autographer and accelerometer. Appendix A displays agreement criteria on an item-by-item basis. This documents the extent to which the reference methods infer agreement was met with the self-reported behaviour. At the population level, boundaries were created to evaluate what is an acceptable percentage agreement to accept a self-reported behaviour to be valid. Good validity was classified as percentage agreement of between 85% and 100%, acceptable validity between 60% and 84%, and poor validity 59% and below. This was a subjective method which was self-determined but was employed to try and define the extent to which health and lifestyle behaviour are deemed valid against the reference methods.

All accelerometer data were converted to minutes. To examine the systematic error and offer a visual representation of between-measure agreement, Bland–Altman limits of agreement (LOA) were used for all CHAT items that provided continuous data [46]. The mean difference (bias) was estimated using paired *t*-tests and 95% LOA were calculated as the (standard deviation (SD) * ± 1.96) for the upper and lower limit, respectively. For items 4b/c and 11b/c (See Table 1), the autographer was taken as the reference method. For some items, data points were excluded from the analysis because participants did not comply with the protocol. For instance, if a participant did not wear the autographer and the accelerometer during a part of the day, they were excluded from the analysis. All items were analysed twice using percentage (%) agreement ((Number of participants in agreement) / (sample size × 100)) [47] to compare self-reports to each reference method, respectively.

The transcripts gathered from commentary and interviews were inductively analysed to explore emergent themes surrounding the discussed topics. Pen profiling (41) was used to present data stratified by sex and respondent type. The sample size (*n*) displayed for each theme reflects the individual responses. This process remained flexible, with commonly cited themes recorded until data saturation was reached [48]. Triangulation occurred by presenting profiles alongside verbatim materials, which were cross-examined by the first author. This was to ensure methodological rigour and achieve dependability [49,50]. Coding was used to present raw transcripts as follows: B = boy, G = Girl, M = mother, and F = father.

## 3. Results

Participants responded well to wearing devices in synchronisation, with the autographer and accelerometer worn throughout the day. A total of 25,966 images were collected across the 14 participants. A total of 25,962 images were screened, including 9073 that were excluded from analysis as they were deemed unusable, for example, as a result of the lens being closed, images obscured by an object and/or body part, and the autographer being removed to show friends. Instances of deactivation and removal occurred due to bathroom use, engaging in vigorous activities and the autographer battery running low. The average wear-time was 10 h (h) 36 min (min; ±3 h 16 min), with the average placement time of 07:37 (±0:27 h:min) and removal at 18:17 (±3 h 15 min). Ten out of 13 (77%) participants fully documented reasons for removal and replacement of devices, although one participant failed to return the booklet. Four images were removed by parents and one by the researcher during data screening. Images were deleted due to religious reasons (*n* = 3), explicit images (*n* = 1), and parent unease (*n* = 1).

### 3.1. Method Comparison

A summary of percentage agreements from CHAT items versus the autographer and accelerometer, for items associated with time, are shown in Table 2 and Table 3, respectively. Bland–Altman plots were performed for six items associated with time (Figure 1, Figure 2, Figure 3, Figure 4, Figure 5 and Figure 6). Results showed participants produced good validity in the following seven items: Getting up time, lunch type, mode and context of travel to and from school, and oral hygiene. Acceptable validity was obtained in the following five items: Sleep time, breakfast type, before and after school time spent completing homework, and lunchtime physical activity intensity. Poor validity was achieved in the remaining nine items: Before and after school screen time, breakfast and lunch drink type, fruit and vegetable servings, before and after school time spent doing sports and exercise, and morning and afternoon break physical activity intensity.

#### Interview Analysis

One father contacted the main researcher with concerns over the use of the autographer after receiving the information sheet. These related to the legality of undertaking the research. This parent perceived the camera as an invasion of privacy:
“I find the consent form and your study, which would appear to involve a camera randomly taking pictures, gross violation of privacy dressed up as something cool to be part of because it’s the newest (privacy invading) technology.”

The parent then went on to query if the researchers had obtained permission from the police to undertake the research:
“It’s obvious the consent form is there to get around any legal issues you could face for such pictures, but I would like to ask if you had permission from the police to take random pictures of school children at the weekend, in their bedrooms getting changed etc., with or without consent?”

The parent closed their comments by stating others had similar concerns. However, no evidence was produced to support this statement:
“I have spoken to several other parents this morning, who all feel exactly the same, meaning they do not want you taking pictures of them randomly. Would you allow the children to give you a camera for the weekend to take random pictures of your life?”

Figure 7 illustrates reasons for participating and reactions to receiving the information sheet. Four participants gave their reasons for volunteering. The most common theme was from two boys and one girl who were interested to find out about their lifestyle behaviours:
“To gauge my activities.”(B10)

Other reasons demonstrated children were driven and excited by the opportunity to wear a novel device.
“I was excited to wear a camera, and yeah, that it could see what you were doing and how long you were spending in certain places of certain activities.”(B14)

Finally, one child wanted to take part as some of their peer group were involved.
“I dunno, I saw people doing it and I thought it would be like cool to do it.”(B12)

Responses showed two dominant themes when children were asked how they felt wearing the autographer. The first was that it was a positive experience, which was expressed by twelve children (nine boys, three girls), Three children conceptualised using phrases as:
“It was fun using the camera.”(B10)
“I enjoyed it.”(G6)

Others demonstrated that they felt *“cool”* and *“special*”.

Six participants reported an increase in attention received from other pupils. A subtheme emerged which involved peers wanting to have their photographs taken. This was mentioned by four children. Figure 8 depicts responses on how participants felt at home and in public. The most frequent theme related to the children’s interaction with others. Four boys reported that their family did not want to be caught on camera, whereas one girl reported their siblings enjoyed being filmed, while another said their siblings paid no attention. Three boys and one girl noted family and friends being intrigued by the autographer. One boy mentioned being approached by a member of the public.
“At the park I had a problem. The guy was asking me if the camera was videoing. I said it was just recording photographs. I told him it just took pictures and then walked away.”(B14)

There were some negative responses about wearing the device, with one mother and one boy saying that they felt self-conscious in public.

There were several explanations given for removing or not wearing the autographer all the time (five responses). Some children simply forgot to put the camera back on, while others mentioned the battery ran low. Other children took it off due to an argument with a family member. This theme was mirrored in parental responses, with two fathers and three mothers commenting on the poor battery life (Figure 9).

The next cited was concern over privacy, which was mentioned by four mothers, with one vehemently worried by the impact the autographer had in the home environment.
“My only concerns were the camera recording the sound and you hearing how much I shout. And then his younger brother being silly around the camera, I can’t remember which day but his younger brother getting his bum out.”(M11)
“The camera invades privacy. Dangerous when people are coming out of their showers in the morning, getting dressed, etc. In particular a problem as G5 gets dressed before the rest of us and her siblings. So, her older sister was mainly stressing, walking around in the morning without being caught by the camera. G5 did end up taking the camera off during the weekend day because she got angry with it, after she’d had an argument. We have younger children you see, so, members of our family did not want to be seen eating their dinner, etc., normal daily tasks were invaded. I found the camera caused lots of arguments due to people not wanting to be seen in their underwear or eating dinner. Not wanting to be photographed.”(M5)

Three parents were worried about the lanyard. Two comments referred to the camera moving excessively when undertaking daily tasks.
“Improvement could have been made to the lanyard as it kept swinging around and would face the ceiling, face the wrong way. Then B12 would have perhaps felt more comfortable wearing it more for his activities throughout the day.”(F12)

One father and one mother made reference to family members not wanting to be filmed.
“My husband would make B8 avoid him being filmed. So B8 would creep around the room trying to not get this dad in the shot.”(M8)

Finally, one comment was made by a mother who was worried that their child might damage the equipment and that the autographer did not capture all instances of physical activity.
“You weren’t getting a true reading of his day. Whereas, he doesn’t necessarily do physical activity during the day, but there were parts where he was and they weren’t being recorded because he was too frightened to have it on in case it broke.”(M13)

Two mothers and one parent–child dyad suggested changing how the device was worn. Many commented that the lanyard should be replaced. Six respondents stated that the improvements to the lanyard should focus on stabilising and securing the autographer to maintain a fixed position. The overall consensus was that if the autographer lanyard was secure, then children could wear it during physical activity. Suggestions to change the buttons on the device were raised by two mothers and one child. They said that they were unable to ascertain whether the autographer was on or off. Comments to improve battery life were also suggested.

Overall, four children agreed they felt the experience was positive. One summarised his experience as:
“Easy and fun”(B2)

Two parents provided comments.
“It’s been interesting watching his day in a snapshot. You know, stuff that I don’t see you know as well in the playground. I think it’s been great. We are more aware of the amount of screen time.”(M11)
“I think it’s been positive, and I would recommend anybody to do it.”(M7)

## 4. Discussion

There were two aims to this study. The first was to validate 21 items regarding self-reported health and lifestyle behaviours in 9–11-year-old children using CHAT and objective measures derived from an autographer and accelerometer. Second, the ethical challenges of taking regular photographs using an autographer were evaluated through interviews with children and their families.

Findings indicate that children can accurately report on some health and lifestyle behaviours using the CHAT. Overall, seven, five, and nine items produced good, acceptable, and poor validity, respectively. The most promising findings were how accurately children reported mode and context of travel to and from school. The use of both the accelerometer and the autographer to classify the physical activity intensity and the context of the journey was invaluable. This is a good example of where using two objective methods to validate self-reported behaviour is very beneficial to classify children’s health and lifestyle behaviours, and to date, this is the first study to use this methodology. Among other questions most accurately recalled were self-reported breakfast and lunch type, showing acceptable and good validity, respectively. Comparing these findings to others, previous research has produced mixed findings when verifying self-reported breakfast and lunch type [51,52,53]. This discrepancy in recall accuracy is likely attributed to web-based methods being less cumbersome and more interactive and motivational, which is an important factor in conducting dietary recall [54]. Another interesting finding is that lunch was more accurately reported than breakfast, which could be explained by the greater time between eating and reporting breakfast than lunch. In dietary recall, this phenomenon termed retention interval has been evidenced elsewhere [55]. Other behaviours which agreed favourably were getting up time, which 12 participants recorded accurately. This is a useful behaviour to self-report, as it enables this information to be gathered that otherwise might not be obtained with parental proxy reports.

Self-reported screen-based behaviours and time spent completing homework produced mixed findings. This was because items related to screen time showed poor validity, whereas self-reported homework achieved acceptable validity. One conclusion that could be drawn from these findings is that children are more accurate at reporting behaviours over a smaller time period, which is in accordance with exploratory studies [43,44]. This is conceptualised by before-school screen time being reported more accurately than after school screen time. Together, both these findings suggest higher validity can be achieved if smaller time frames are used when constructing items for questionnaires. However, time spent in homework remained similarly valid for before and after school, which might suggest the salience of an event and the detail of recall can influence recall accuracy [56]. For example, many children view homework as a task that has regular prompts and reminders from teachers and parents, which result in the event being more memorable—as opposed to screen time, which is a regularly occurring behaviour among school children.

When relating results to question type, participants gave higher accuracy for items requiring categorical selection, such as active transportation and breakfast consumption. For questions requiring participants to estimate duration and time, often poor validity was obtained. This is supported by research which shows children face difficulty estimating time [43], due to the cognitive burden of recall. Further, children were more likely to underestimate screen time, perhaps due to the larger time frame and the precision of questions in the CHAT. This may relate to technological advancements, where several electronic items that are used by children may not have been included in the CHAT question. This could thereby preclude their ability to determine the total accumulated screen time. Indeed, research has highlighted that the speed of technological advancements is surpassing methods for assessing screen time [57]. Hence, while results partially support previous research showing children struggle to estimate time spent in behaviours [58], an awareness of the prominence of an event to a child, alongside an up-to-date understanding of current advancements and norms relating to each health behaviour, is essential in eliciting accurate responses in children of this age. In the current findings, poor validity was obtained for both before- and after school screen time, showing children could not sufficiently report accurately screen time engaged in.

An area in which the CHAT self-report consistently showed poor validity was time spent engaged in sports or exercise before, during (morning and afternoon break) and after school. This can be directly linked back to other researchers who have also demonstrated this [58]. However, self-reported time spent after school doing sports and exercise showed contradictory findings. The accelerometer and autographer achieved 54% and 73% agreement, respectively. This could be due to the autographer distinguishing between sedentary behaviour and physical activity by providing context-specific information. Conversely, hip-worn accelerometers cannot identify posture nor certain physical activities, such as lower body exercises [59]. Nonetheless, wearable cameras could also be impeded by the inability to wear for water-based activities or contact sports. This could explain, at least in part, the discrepancy as associated with the poor validity found for this question. A major advantage of combining both accelerometry and the autographer is providing context across the entire day, which remains understudied in children. This study therefore shows promise for future research, not least because the autographer can infer context and the accelerometer can confirm the intensity of a given activity, which is both novel and invaluable as a methodology.

This study sought to evaluate ethical challenges associated with taking regular photographs using a wearable camera through interviews with children and their families. Results showed that the initial reaction of parents to receiving the study information sheet was that they thought the camera would encroach on their privacy. Children expressed apprehension about being responsible for the device but overall were excited by the technology. However, when children reflected, the majority reported that they found the experience fun and enjoyable. This aligns with a study performed on 11–13-year-olds who also demonstrated the same experiences [24]. The consensus amongst parent–child dyads was that the lanyard needs to be improved to stabilise the autographer, so that it can be worn for a wider range of physical activity. This was expressed as parents felt some important activities and behaviours were missed because of the removal of the device for vigorous and water-based activities as well as for privacy. This is an interesting finding, since this concurs with the conclusions formed by the research team that the autographer was unable to capture all physical activity and sports participation. Similarly, limited battery life was consistently reported as a barrier to full data capture, meaning photographs of some behaviour could not be captured. This mainly concerned behaviours in the evening, such as after school time spent doing screen-based activities and homework, as well the time children went to bed. These methodological limitations reduced the sample size used for some items when validating self-reports. However, a major strengthening point of this study is that the autographer can objectively measure behaviours that cannot otherwise be obtained with other forms of objective measures, such as accelerometers. For instance, time spent completing homework, or number of times a child brushes their teeth. Previous research has mainly used children’s self-reports largely verified by proxy measures from parents [28,31]. The use of wearable cameras combined with accelerometers not only improves the objectivity of measures of physical activity but helps establish the validity of children’s self-report of lifestyle behaviours. Therefore, this study has further progressed findings using a wearable camera to objectively assess health and lifestyle behaviours, which can contribute to the growing body of literature using cameras to investigate a wide array of lifestyle behaviours [19,60,61,62]. Future research should seek to use wearable cameras alongside other objective measures to validate self-reported health and lifestyle behaviours.

In accord with the secondary aim, this study highlighted the importance of adhering to the ethical framework of Kelly et al. [39]. This framework was informed by two research councils; the British Sociological Association and the Economic and Social Research Council. A major theme of the guidelines is that they are centred on participant informed consent. This corresponds to how data are collected and used. Four principles of ethics are stated: (I) *Respect for autonomy*; related to informed consent and confidentiality; (II) *beneficence*; responsibility to do well; (III) *nonmaleficence*; obligation to avoid harm; and (IV) *justice*; the importance of benefits being equal to the burden of research. These principles have manifested from approaches used in public health research. The current study was conducted ensuring parent–child dyads were integrated in the data collection and screening process by allowing them to view images in private. If the concerns of parents and children are targeted at the initial recruitment stage, this will alleviate the scepticism around this device, enhancing recruitment and compliance—in particular, educating participants on why the use of the camera is beneficial to explore health and lifestyle behaviours. Rather than being wary of it, they can understand its purpose. This, in turn, will increase study sample size and participant retention. The present study employed this approach to ensure every step of the process was open and participants understood the wearable camera and why it was being used. Further studies are needed to explore the acceptability and barriers of wearable cameras, similar to the study undertaken by Cowburn et al. [63]. In future, if better knowledge is obtained on perceptions of wearable cameras by children and parents, this will help to tailor recruitment strategies as well as inform how wearable cameras can be used to their full potential, to investigate children’s health-related behaviour.

Despite limited findings in this study, the mixed-methodology design was a novel and plausible method to be reproduced in future using wearable cameras in combination with an accelerometer. As evidenced, the use of the camera technology presents unique methodological and ethical challenges. Improvements in autographer placement, such as fixing it to clothing, will enable a more reliable capturing of all forms of physical activity and provide a better representation across the whole day. Behaviours that parents otherwise may not otherwise be aware of, such as during school hours, can be observed. Further exploration on how the battery life of the device can be extended to ensure the whole day is captured is needed, while ensuring photograph quality is not hindered. This study’s findings can also contribute to the development of ethical guidelines and recruitment approaches that are tailored for children using wearable cameras. Based on a mixed-methodological approach, it was clear the autographer has potential as a practical method capturing behaviours that cannot otherwise be obtained using self-report [18,62]. In accord with the primary aim of the study, it is clear that children could provide valid responses to some health and lifestyle behaviours but not all. Findings show participants produced good, acceptable, and poor validity in seven, five, and nine items, respectively.

While the study findings are positive, it is important to clarify the limitations. Firstly, the sample was limited in size and ethnic, social, and cultural diversity. Moreover, all the participants were aged between ten and 11 years, thereby limiting the validation of self-reports to these target ages. Nonetheless, the ability to take photographs and observe all areas of a child’s lifestyle behaviour was invaluable. On the other hand, data processing was burdensome and time-consuming. In light of this, studies have now explored the used of image-analysis software to make processing of images more efficient [64], as well as using ecological momentary assessment for assessment of physical activity [65]. Therefore, future research can begin to combine analytical techniques to reduce the data processing burden associated with wearable cameras.

## 5. Conclusions

The present study demonstrated that the current mixed-method design was able to validate some items included in the CHAT using an accelerometer and autographer. This is evidenced with seven, five, and nine CHAT items showing good, acceptable, and poor validity, respectively. Despite this, the autographer shows potential to improve concurrent validity compared to parent proxy reports when used alongside accelerometers. However, it is clear that wearable cameras have some technical and ethical challenges, such as privacy. Nonetheless, not all participants had this view and others received the device positively. Future research using wearable cameras needs to work within an ethical framework in conjunction with a family-orientated approach. Further, results clearly indicate that future research must be completed to improve the autographer functionality and battery life to offer a better representation of the distribution of physical activity across the entire day. Once wearable camera functions are optimised, they could overcome the need to use parental proxy reports, conceptualising behaviours that parents would otherwise be unaware of. Based on the items shown to be valid, CHAT has been revised to incorporate more categorical questions and less time-focused questions, thereby further progressing its development as a self-report tool.

## Figures and Tables

**Figure 1 children-06-00020-f001:**
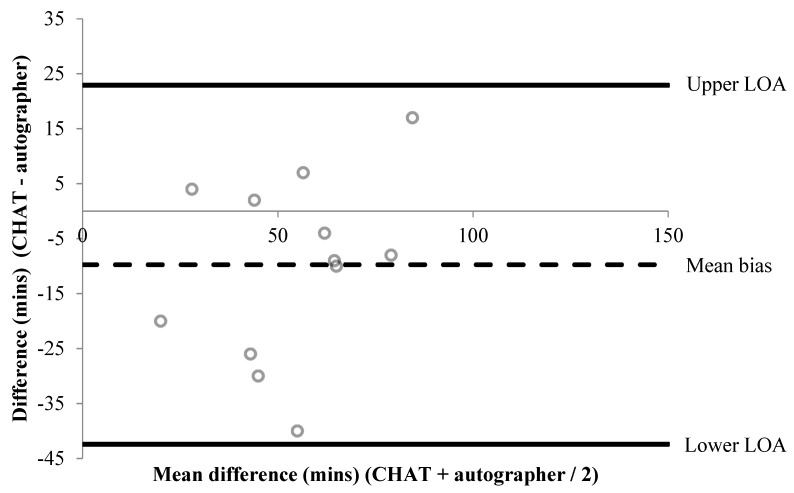
A Bland–Altman plot demonstrating the difference between self-reported and autographer-derived getting up time (*n* = 12). A mean bias of −10 min (±17 min) with limits of agreement (LOA) of +23 and −42 min is shown.

**Figure 2 children-06-00020-f002:**
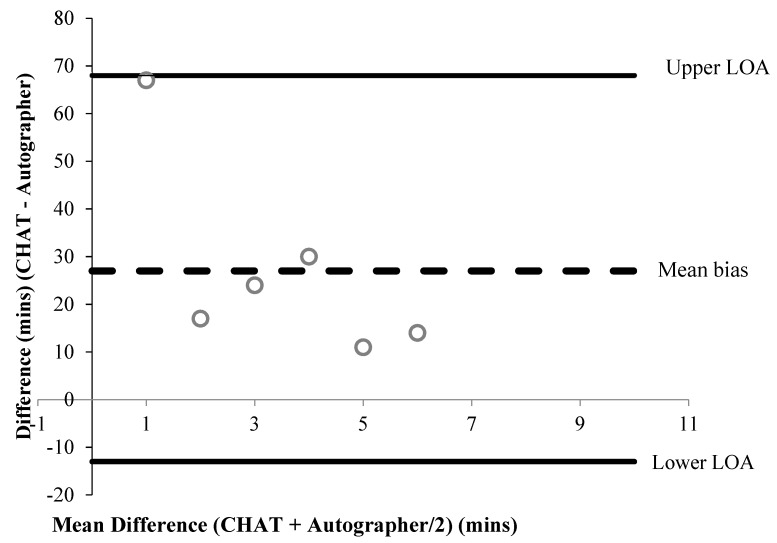
A Bland–Altman plot demonstrating the difference between self-reported and autographer-derived sleep time (*n* = 6). A mean bias of +27 min (±21 min) with LOA of + 68 and −13 min is shown.

**Figure 3 children-06-00020-f003:**
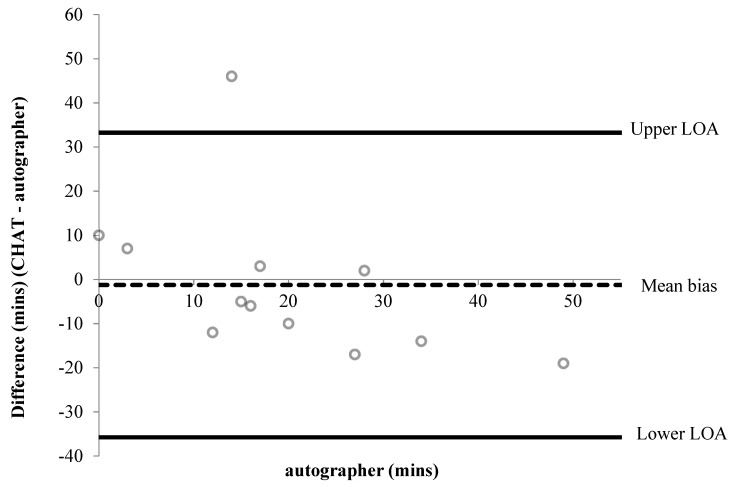
Bland–Altman plot demonstrating the difference between self-reported and autographer-derived before lessons screen time (*n* = 12). A mean bias of −1 min (±18 min) with LOA of + 33 and −36 min is shown.

**Figure 4 children-06-00020-f004:**
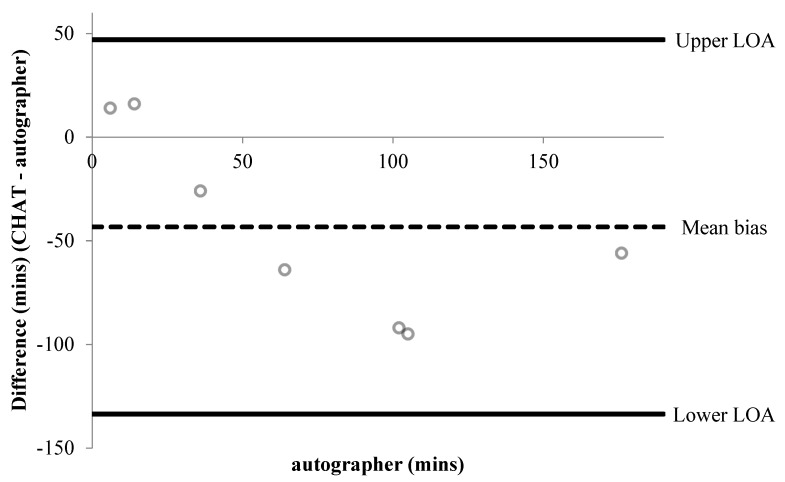
Bland–Altman plot demonstrating the difference between self-reported and autographer-derived after school screen time (*n* = 7). A mean bias of −43 min (±46 min) with LOA of +47 and −134 min is shown.

**Figure 5 children-06-00020-f005:**
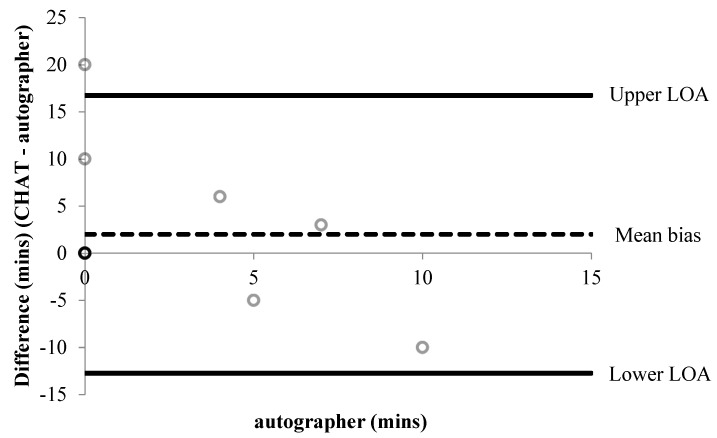
Bland–Altman plot demonstrating the difference between self-reported and autographer-derived homework and reading duration before school (*n* = 12). Six data points overlap shown by the darker outline. A mean bias of +2 min (±8 min) with LOA of +17 and −13 min is shown.

**Figure 6 children-06-00020-f006:**
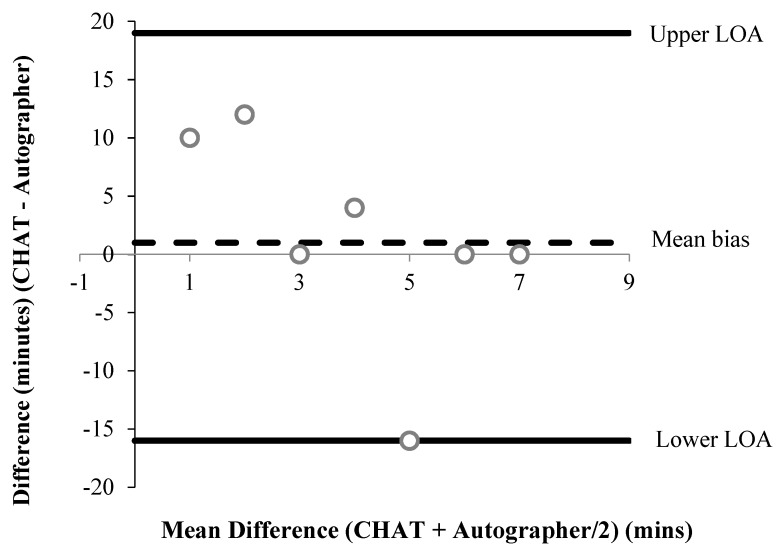
Bland–Altman plot demonstrating the difference between self-reported and autographer-derived homework and reading duration after school (*n* = 7). A mean bias of +1 min (±9 min) with LOA of +19 and −16 min is shown.

**Figure 7 children-06-00020-f007:**
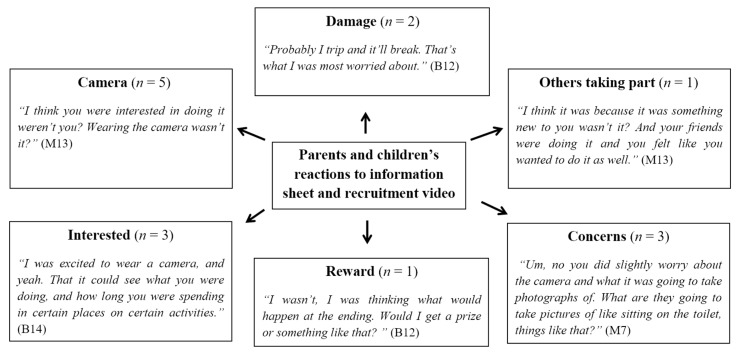
Pen Profile showing parents’ and children’s reactions to the information sheet and the recruitment video (B = Boy, G = Girl, F = Father, M = Mother); *n* shows the number of individual responses.

**Figure 8 children-06-00020-f008:**
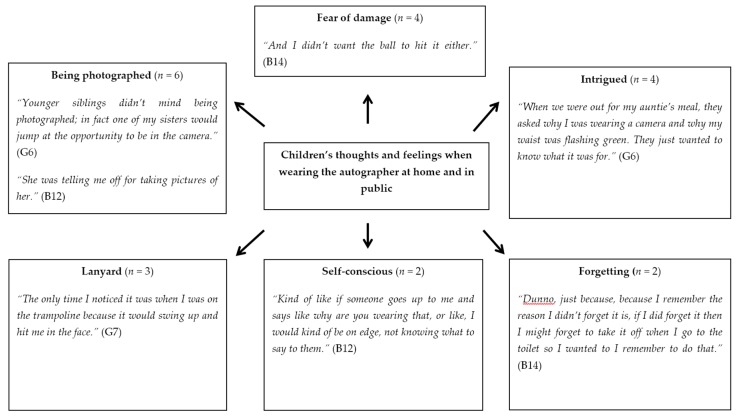
Pen profile depicting children’s thoughts and feelings when wearing the autographer at home and in public (B = Boy, G = Girl). *n* shows the number of individual responses.

**Figure 9 children-06-00020-f009:**
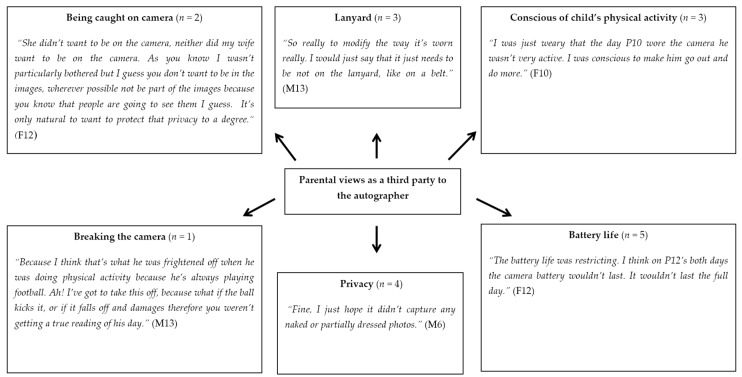
Pen profile illustrating parental views as a third party to the autographer (F = Father, M = Mother). *n* shows the number of individual responses.

**Table 1 children-06-00020-t001:** Summary of twenty-one items tested for concurrent validity (questions 1–14).

Item	Question
1	What time did you get up yesterday?
2	What did you eat for breakfast yesterday?
3	What did you drink for breakfast yesterday?
4a	Before lessons started, how long did you spend doing sports or exercise?
4b	Before lessons started, how long did you spend sitting down watching TV/playing video games/using iPad/internet?
4c	Before lessons started, how long did you spent doing homework or reading?
5a	How did you get to school?
5b	Did you travel with an adult?
6	What did you do for most of your morning break?
7a	What did you have to eat for lunch?
7b	What did you have to drink for lunch?
8	Apart from eat your food, what did you do for most of your lunchtime break?
9	What did you do for most of your afternoon break?
10a	How did you travel home from school?
10b	Did you travel with an adult?
11a	After school, how long did you spend doing sports or exercise?
11b	After school, how long did you spend sitting down watching TV, playing video games/using iPad/internet?
11c	After school, how long did you spend doing homework or reading?
12	How many portions of fruit and veg did you eat yesterday?
13	How many times did you brush your teeth yesterday?
14	What time did you go to sleep?

**Table 2 children-06-00020-t002:** Percentage agreement for between-measure agreement of self-reports compared to the autographer for all 21 items.

Question	*n*	Agreement	% Agreement
Before lessons started, how long did you spend doing sports or exercise?	12	5	42
After school how long did you spend doing sports or exercise?	11	8	73
What did you have for breakfast yesterday?	12	9	75
What did you do for most of your morning break intensity?	13	7	54
What did you have to eat for lunch?	14	14	100
What did you do for most of your lunchtime?	13	8	62
What did you do for most of your afternoon break?	13	5	38
How did you get to school yesterday morning?	14	13	93
Did you travel with an adult?	14	13	93
How did you get home yesterday?	10	9	90
Did you travel with an adult?	10	9	90
What did you drink for breakfast yesterday?	12	3	25
What did you drink for lunch yesterday?	14	8	57
How many portions of fruit and vegetables?	12	6	50
How many times did you brush your teeth yesterday?	10	10	100
What time did you get up yesterday?	12	11	93
What time did you go to sleep?	6	5	83
Before-school screen time	12	6	50
After school screen time	7	0	0
Before-school time spent on homework	12	8	67
After school time spent on homework	7	5	71

**Table 3 children-06-00020-t003:** Percentage agreement for physical activity and active transport questions when comparing self-reports to accelerometry.

Question	*n*	Agreement	% Agreement
Before lessons started, how long did you spend doing sports or exercise?	12	4	33
What did you do for most of your morning break yesterday?	14	8	57
What did you do for most of your lunchtime?	14	10	71
What did you do for most of your afternoon break?	14	6	46
After school how long did you spend doing sports or exercise?	13	7	54
How did you get to school yesterday morning?	14	13	93
How did you get home yesterday?	10	11	91

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
