# Peer review of "Can Wearable Cameras Be Used to Validate School-Aged Children’s Lifestyle Behaviours?"

_children, 2019, doi:10.3390/children6020020_

Round 1
Reviewer 1 Report
It would be good to indicate how the sample is chosen and what procedures are followed. The analysis of the interviews is not clear, it would be advisable to improve the presentation format
Author Response
Comment 1: It would be good to indicate how the sample is chosen and what procedures are followed.
Reply: More detail provided in methods section; subheaded participants Page 2, line 84
Comment 2: The analysis of the interviews is not clear.
Reply: Page 6, line 227 to 229. The transcripts gathered from commentary and interviews were inductively analysed to explore emergent themes surrounding the discussed topics. Pen profiling (41) was used to present data stratified by sex and respondent type.
Comment 3: It would be advisable to improve the presentation format.
Reply: Presentation updated throughout
Reviewer 2 Report
This article aims to investigate the validity of a selection of items from a questionnaire (Child Health and Activity Tool; CHAT) which assesses health behaviours of children and address the ethical and privacy concerns of using wearable cameras in this population. The main finding, presented by the authors, is that wearable cameras are feasible to use in this population and that the CHAT has validity for some health behaviours but not all.
While the authors should be commended for their novel approach to measurement of health behaviours in children, I believe there are several aspects of the paper that require major revision and I have provided some comments for consideration below.
1. It is unclear in what respect the authors feel wearable cameras are feasible. Considering the poor response rate, the subsequent small and homogenous sample, the high cost, the mostly negative views from both children and parents, and the lack of agreement with activity behaviours (other than mode and timings), I think it is inaccurate to state that wearable cameras are feasible for use with school-aged children. I also think it is misleading to say in the title of the paper that wearable cameras can validate lifestyle behaviours in light of the modest agreement across items of the questionnaire (particularly where responses require a level of quantification).
2. The rationale for the study is a little confusing – the authors are suggesting that a web-based questionnaire is better than accelerometers, fitness tests and diaries, and indeed the main aim is to validate a questionnaire against accelerometery, but I’m not sure why the use of wearable cameras (which I assume are also costly, require high participant burden and have obvious ethical concerns) are being justified for use in measuring behaviours. It seems a little contradictory.
3. The CHAT is mentioned in the aim but is not introduced prior to this. It would be useful to understand why you are using the questionnaire, how it has been developed, whether it is web-based or not, how it is administered and why you are validating it.
4. Why were only certain items from the questionnaire chosen to be validated? The authors do not provide a reason for the selection of these items or why the other seemingly relevant 15 items were excluded. Furthermore, 14 is a very small sample size for a validation study, therefore I do not see how the results can offer any conclusion about validity or use of the questionnaire as a whole in the target population.
5. It is unclear what the final sample size was for analysis. The authors mention a sample of 14, but that some were required to repeat data collection due to non-wear time (and that some refused) and later in the discussion it is mentioned that sample size was reduced with respect to validation of the CHAT. Please clearly state the N value for each result reported.
6. Please provide a brief explanation of the wear time criteria used. It is mentioned at line xxx that a minimum of 8 h wear time was required, yet only the school day (presumably 6 h?) was used in analysis. Can you clarify why you required 8 h of wear time and also how many particiapnts were included in final analysis, as you state all 14 provided valid data but then some were asked to repeat the – was this repeat wear coincide with all study outcomes (i.e. a repeat of the questionnaire on the day following the repeated accelerometer wear?
7. I’d like to suggest a clearer reporting of the Bland-Altman analysis used in this study and direct the authors to the following papers which may be of use - Chhapola, V., Kanwal, S. K., & Brar, R. (2015). Reporting standards for Bland–Altman agreement analysis in laboratory research: a cross-sectional survey of current practice. Annals of Clinical Biochemistry, 52(3), 382–386. https://doi.org/10.1177/0004563214553438; Abu-Arafeh, A., Jordan, H. & Drummond, G. (2016). Reporting of method comparison studies: a review of advice, an assessment of current practice, and specific suggestions for future reports. British Journal of Anaesthesia, 117(5), 569-575.
8. Regarding agreement criteria, how have you defined the intensity of different physical activities for this population? For example, if the self-report answer to the question “how did you get to school” was “walked” are you assuming that walking is low intensity (<3 METs) or moderate intensity (3-6 METs) or other? Also, it is difficult to see which criteria relate to which CHAT item in the appendix table – could this be improved?
9. Could the authors discuss the limitations of the actigraph device in measuring sedentary behaviour, i.e. no identification of posture which is an important aspect of the sedentary behaviour definition? This is particularly important considering it is being used for validation purposes.
10. I am unsure how appropriate the use of the term ‘objective’ is when referring to the autographer and actigraph – these are not entirely objective as they can be removed and tampered with to alter results – indeed you have reported that do not accurately capture all activities. I recommend this article which discusses aspects of accelerometer-based measurement of physical activity behaviours - Pedišić Ž, Bauman A. Accelerometer-based measures in physical activity surveillance: current practices and issues. Br J Sports Med 2015;49:219-223. You may wish to consider the term ‘device-based’ measurement.
11. The opening line of the conclusion states “Overall, the present study demonstrates that the Autographer is a valid measurement tool to assess a wide range of health-related behaviours in children.” Firstly, the Autographer was not what was being validated. Secondly, to say a “wide-range of health behaviours” is not accurate. The second paragraph of the conclusion states “Children and their families’, perceptions of using the Autographer was positive.” This is not my interpretation of the results which appear to report largely negative feedback.
Minor comments:
Overall, I’d recommend proof-reading for spelling and grammatical errors as well as clarity in writing. The manuscript is a little difficult to read at points. The manuscript appears to be an early draft with missing references (see Page 10 line 363 and Page 11 line 420).
Page 2 line 63 - could you provide a very explanation of what the Autographer is? It only becomes apparent in later sections of the paper.
Page 3 line 99 - What is a medium capture rate? How many images?
Page 3 line 101 – do you mean mid axillary line?
Page 5 line 179 – Do you mean parent-child dyad?
Page 9 line 281 – Fathers not fathered
Page 11 lines 386-389 – please clarify the meaning of this sentence.
Page 11 line 413- Can you explain what the framework includes?
Page 11 line 421 and 426, page 12 443 – what promising/positive findings are you referring to? So far, much of what you have described regarding the wearable camera (other than a small number of children finding it enjoyable) is negative and with respect to the validity of [part of] the questionnaire only certain items showed acceptable-good agreement.
Page 12 – avoid repeating results.
Page 12 line 436 – you refer to seven items having good validity here but eight elsewhere.
Author Response
Comment 1: It is unclear in what respect the authors feel wearable cameras are feasible. Considering the poor response rate, the subsequent small and homogenous sample, the high cost, the mostly negative views from both children and parents, and the lack of agreement with activity behaviours (other than mode and timings), I think it is inaccurate to state that wearable cameras are feasible for use with school-aged children. I also think it is misleading to say in the title of the paper that wearable cameras can validate lifestyle behaviours in light of the modest agreement across items of the questionnaire (particularly where responses require a level of quantification).
Reply: We embarked on this research to overcome the problem with the use parental reporting to validate children’s self reporting of their behaviours. This was the first study to do so and brings into question all previous approaches in the literature. That said the study is small scale so cannot generalise but the strength is in the message. First cameras can be used to assess children’s behaviours, second that they are not perfect and carry some ethical and technical challenges but third the results they produce has the “potential” to improve concurrent validity compared to parent report.
Comment 2: The rationale for the study is a little confusing – the authors are suggesting that a web-based questionnaire is better than accelerometers, fitness tests and diaries, and indeed the main aim is to validate a questionnaire against accelerometry, but I’m not sure why the use of wearable cameras (which I assume are also costly, require high participant burden and have obvious ethical concerns) are being justified for use in measuring behaviours. It seems a little contradictory.
Reply: The literature has been supportive that the cameras can capture in field behaviours. We were also interested in physical activity and thus aligned the camera data to accelerometer data and of course self report using the CHAT. Given the number of valid photographs taken and how these contextualised behaviour the cameras were accurate in recording behaviours that parents would not be aware of
Comment 3: The CHAT is mentioned in the aim but is not introduced prior to this. It would be useful to understand why you are using the questionnaire, how it has been developed, whether it is web-based or not, how it is administered and why you are validating it.
Reply: See page 2 – background of where and how the CHAT was developed. CHAT is an example of a web based questionnaire developed and designed for primary school children to produce a quick, easy method of gathering data on an array of health behaviours in 9-11 year old’s. It was established from a paper based questionnaire as part of SportLinx; a health and fitness initiative for primary school children. Items include wellbeing, sleep, nutrition and physical activity. It can be completed on computer, laptop or iPad and is administered in the primary school setting in the presence of researchers. It has not been validated before, this needs to be addressed.
Comment 4: Why were only certain items from the questionnaire chosen to be validated? The authors do not provide a reason for the selection of these items or why the other seemingly relevant 15 items were excluded. Furthermore, 14 is a very small sample size for a validation study, therefore I do not see how the results can offer any conclusion about validity or use of the questionnaire as a whole in the target population.
Reply: The reason only 21 of the items were validated was because the latter section of the questionnaire related to child happiness with health, fitness, friends, family and school, as well as general autonomy and competence. Also, part of the questionnaire referred to a weekly recall of health and lifestyle behaviours. Assessing weekly behaviour would have been very burdensome on children of this age. Consequently, to make the currently study feasible, only items pertaining to previous day recall and health questions were validated. The research team were limited by time constraints, budget and equipment. Also, the challenging nature of the novel mixed methodology resulted in difficulty recruiting. This is why several different avenues were explored to recruit and retain compliance. The overall sample size is a reflection of this.
Comment 5: It is unclear what the final sample size was for analysis. The authors mention a sample of 14, but that some were required to repeat data collection due to non-wear time (and that some refused) and later in the discussion it is mentioned that sample size was reduced with respect to validation of the CHAT. Please clearly state the N value for each result reported.
Reply: See page 4, line 131 to 143. Fourteen children were recruited in the study. Of these, nine met the 8 hour wear time criteria at the first observations. Of the remainder, two agreed to rewear and met the wear time criteria at the second observations, and three declined to rewear. All fourteen data sets were used for analysis. However, if either device were not worn for the given time frame relevent to the question, then then the data was excluded in the analysis. This is shown in Table 1 with sample size stated for each item.
Comment 6: Please provide a brief explanation of the wear time criteria used. It is mentioned at line xxx that a minimum of 8 h wear time was required, yet only the school day (presumably 6 h?) was used in analysis. Can you clarify why you required 8 h of wear time and also how many participants were included in final analysis, as you state all 14 provided valid data but then some were asked to repeat the – was this repeat wear coincide with all study outcomes (i.e. a repeat of the questionnaire on the day following the repeated accelerometer wear?
Reply: See page 4, line 128 to 143. An 8 hour period was chosen for wear time as this represented the average battery life during the pilot study obtained using the medium capture rate. This allowed for 6 hours of the school day to be gathered, plus one hour pre and post school. 14 participants recruited in the study. Of these, nine met the 8 hour wear time criteria at the first observations. Of the remainder, two agreed to rewear and met the wear time criteria at the second observations, and three declined to rewear. However the analysis was performed item by item. If a particular individual did not wear either the accelerometer or autographer for given time period pertaining to the question then they were excluded in the analysis. This is shown with sample size varying between each CHAT item. For e.g. the sleep question only had 6 data sets whereas the lunch type had all 14 data sets. This is because the challenges with the autographer battery life and the burden of the devices played a role in how much data was collected by each child.
Comment 7: I’d like to suggest a clearer reporting of the Bland-Altman analysis used in this study and direct the authors to the following papers which may be of use - Chhapola, V., Kanwal, S. K., & Brar, R. (2015). Reporting standards for Bland–Altman agreement analysis in laboratory research: a cross-sectional survey of current practice. Annals of Clinical Biochemistry, 52(3), 382–386. https://doi.org/10.1177/0004563214553438; Abu-Arafeh, A., Jordan, H. & Drummond, G. (2016). Reporting of method comparison studies: a review of advice, an assessment of current practice, and specific suggestions for future reports. British Journal of Anaesthesia, 117(5), 569-575
Reply: The research team are unable to address this as the main author writing the paper does not have access to SPSS statistical software to perform Bland-Altman analysis. We would need further time to access SPSS with the aid of the corresponding author to address this and redo bland Altman analysis.
Comment 8: Regarding agreement criteria, how have you defined the intensity of different physical activities for this population? For example, if the self-report answer to the question “how did you get to school” was “walked” are you assuming that walking is low intensity (<3 METs) or moderate intensity (3-6 METs) or other? Also, it is difficult to see which criteria relate to which CHAT item in the appendix table – could this be improved?
Reply: See page 5, line 184 to 191 AND page 4, 156 to 162. The research team did not use MET’s. PA cut off points determined, sedentary, light, MVPA. Then, context of autographer image combined with accelerometry trace, confirmed either images annotation as sedentary (lying, sitting, standing), walking or moderate. See page 17 – made formatting changes for appendix.
Comment 9: Could the authors discuss the limitations of the actigraph device in measuring sedentary behaviour, i.e. no identification of posture which is an important aspect of the sedentary behaviour definition? This is particularly important considering it is being used for validation purposes.
Reply: See page 12, line 425 to 428. The limitation of hip-worn accelerometers are that they cannot identify posture and hence cannot identify sedentary behaviour. Also, shortcomings of captured physical activity could occur due to non-wear time for water based activities such as swimming as well as other periods of the day. However, the strength of the above study design is that although sedentary behaviour could not be identified with the accelerometer, the autographer could define this. For instance, if a child was sat watching TV afterschool, the accelerometer would show periods of inactivity, which the camera could define as “sitting” on the sofa. Effectively the autographer, addresses this shortcoming in accelerometry based measures.
Comment 10: I am unsure how appropriate the use of the term ‘objective’ is when referring to the autographer and actigraph – these are not entirely objective as they can be removed and tampered with to alter results – indeed you have reported that do not accurately capture all activities. I recommend this article which discusses aspects of accelerometer-based measurement of physical activity behaviours - Pedišić Ž, Bauman A. Accelerometer-based measures in physical activity surveillance: current practices and issues. Br J Sports Med 2015;49:219-223. You may wish to consider the term ‘device-based’ measurement.
Reply: The research team agree that yes any monitor can be taken off by a participant and tampered with, questioning its validity. However, this rarely happens in the literature. All measurement techniques have limitations and flaws. The research team have decided “objective” is better suited to describe the accelerometer and wearable camera as it aligns with physical activity literature and the terminology widely used (see publications below).
1) Trost SG, O'neil M. Clinical use of objective measures of physical activity. Br J Sports Med. 2014 Feb 1;48(3):178-81.
1) Farooq MA, Parkinson KN, Adamson AJ, Pearce MS, Reilly JK, Hughes AR, Janssen X, Basterfield L, Reilly JJ. Timing of the decline in physical activity in childhood and adolescence: Gateshead Millennium Cohort Study. Br J Sports Med. 2018 Aug 1;52(15):1002-6.
2) King AC, Parkinson KN, Adamson AJ, Murray L, Besson H, Reilly JJ, Basterfield L, Gateshead Millennium Study Core Team. Correlates of objectively measured physical activity and sedentary behaviour in English children. The European Journal of Public Health. 2010 Jul 22;21(4):424-31.
3) Pearce MS, Basterfield L, Mann KD, Parkinson KN, Adamson AJ. Early predictors of objectively measured physical activity and sedentary behaviour in 8–10 year old children: the Gateshead Millennium Study. PLoS One. 2012 Jun 20;7(6):e37975.
Comment 11: The opening line of the conclusion states “Overall, the present study demonstrates that the Autographer is a valid measurement tool to assess a wide range of health-related behaviours in children.” Firstly, the Autographer was not what was being validated. Secondly, to say a “wide-range of health behaviours” is not accurate. The second paragraph of the conclusion states “Children and their families’, perceptions of using the Autographer was positive.” This is not my interpretation of the results which appear to report largely negative feedback.
Reply: See page 14 – Also, see amended abstract (page 1) and aspects of the discussion. In terms of the perceptions of the autographer, participants and their families had positive and some negative comments. Yes, it was burdensome, in some situations caused some individuals distress, but overall, it was received positively by children and their families. Both sides of the participants’ views should count, as this study was unique in its methodology, and has not been tried before. It was a novel study to provide learning to the research team and others.
Comment 12: Overall, I’d recommend proof-reading for spelling and grammatical errors as well as clarity in writing. The manuscript is a little difficult to read at points. The manuscript appears to be an early draft with missing references. See Page 10 line 363 and Page 11 line 420.
Reply: Missing references – see complete ref list and in text for included references.
Comment 13: Page 2 line 63 - could you provide a very explanation of what the Autographer is? It only becomes apparent in later sections of the paper.
Reply: See page 3, line 115 for introduction of the autographer in the methods. Also, see page 1, line 64 for brief introduction of the autographer.
Comment 14: Page 3 line 99 - What is a medium capture rate? How many images?
Reply: See page 4 > line 120
Comment 15: Page 3 line 101 – do you mean mid axillary line?
Reply: Yes, corrected, page 4, line 125
Comment 16: Page 5 line 179 – Do you mean parent-child dyad?
Reply: Yes, Page 5, line 209
Comment 17: Page 9 line 281 – Fathers not fathered
Reply: Page 9, line 320
Comment 18: Page 11 lines 386-389 – please clarify the meaning of this sentence
Reply: See page 12; lines 421 to 428
Comment 19: Page 11 line 413- Can you explain what the framework includes?
Reply: Page 12, line 467. This framework proposed by Kelly et al (2013) was informed by two research councils; the British Sociological Association and the Economic and Social Research Council. A major theme of the guidelines is that they are centred on participant informed consent. This corresponds to how data are collected and used Four principles of ethics are stated: i) respect for autonomy; related to informed consent and confidentiality; ii) beneficence; responsibility to do well; iii) non-maleficence obligation to avoid harm; iv) justice; the importance of benefits being equal to the burden of research. These principles have manifested from approaches used in public health research.
Comment 20: Page 11 line 421 and 426, page 12 443 – what promising/positive findings are you referring to? So far, much of what you have described regarding the wearable camera (other than a small number of children finding it enjoyable) is negative and with respect to the validity of [part of] the questionnaire only certain items showed acceptable-good agreement.
Reply: See page 13, line 463 to 470. See page 13, line 484. The research team disagree. Both positive and negative views were given by parent-child dyads, and the paper must reflect both viewpoints.
Comment 21: Page 12 – avoid repeating results
Reply: See page 11
Comment 22: Page 12 line 436 – you refer to seven items having good validity here but eight elsewhere.
Reply: Page 13, line 487 – amended in discussion, conclusions and abstract
Reviewer 3 Report
In line 61: the abbreviation CHAT is defined and later the full name appears, line 331.
References appear in parentheses throughout the text, whereas they should appear in square brackets.
From the second reference the repeated numbers appear.
Author Response
Comment 1: In line 61: the abbreviation CHAT is defined and later the full name appears, line 331.
Reply: Amended throughout manuscript.
Comment 2: References appear in parentheses throughout the text, whereas they should appear in square brackets. From the second reference the repeated numbers appear.
Reply: All references chronological in square brackets in text and ordered in reference list with numerical style.
Round 2
Reviewer 2 Report
The following comments have not been addressed satisfactorily:
The Bland-Altman results needs to be reported using Bland-Altman plots and identifying acceptable limits of agreement. This does not require SPSS.
Accelerometer analysis and resultant sample size. Was the 8 h wear time set to identify a valid day of data and if so, where the 8 h wear time was not met, was this data then removed from analysis? It is unclear if the two participants repeating data collection met the wear time criteria on the second attempt. If the remaining three participants did not meet wear time criteria, was this data then removed form analysis? As such, how can Table 2 (which reports agreement with the accelerometer) include data from N=14 participants if at least three did not provide valid data?
In addition:
Lines 372-374 – Please amend. Your results, as presented, suggest children can accurately report SOME health and lifestyle behaviours, not a wide range.
Lines 416-420 – This is confusing, especially line 419. Did the CHAT consistently show poor validity or not?
Lines 429-431 – The paper does not report the distribution of physical activity across the day so how can the authors claim this as novel and invaluable to the study?
Lines 512-516 – this contradicts your abstract and conclusions which now state that your current methodology was not able to validate self-reported behaviours.
Lines 529-532 – this contradicts what is written at lines 372-374
Author Response
Thank you very much for taking the time to re-review the paper. We have responded to your comments on a point-by-point basis below, with all recent revisions in purple text. We believe we have addressed all of your comments and revisions and substantially improved the manuscript.
Comment number
| Reviewer comments
| Author responses |
1. | The Bland-Altman results needs to be reported using Bland-Altman plots and identifying acceptable limits of agreement. This does not require SPSS.
| Thank you for your comment. We agree that these need to be reported and have incorporated the plots and acceptable limits of agreement within the manuscript (see Figures 1-6). |
2. | Accelerometer analysis and resultant sample size. Was the 8 h wear time set to identify a valid day of data and if so, where the 8 h wear time was not met, was this data then removed from analysis?
It is unclear if the two participants repeating data collection met the wear time criteria on the second attempt. If the remaining three participants did not meet wear time criteria, was this data then removed form analysis?
As such, how can Table 2 (which reports agreement with the accelerometer) include data from
| Thank you for highlighting this lack of clarity. Fourteen participants were recruited to participate in the study. Of these, nine met the 8 hour wear-time criteria. However, two (of the five) initially not meeting the wear-time criteria agreed to re-wear the monitors and subsequently met the criteria. Nonetheless, the analysis was performed item by item. If a particular individual did not wear either the accelerometer or autographer for the specific time period in question (i.e. before, during or after school period), then they were excluded from the analysis. For example, only 6 children were included in the “what time did you go to sleep?” question. This meant only children who had the autographer worn up until the time they went to bed, would be used in this analysis. All sample sizes are presented by CHAT item in Tables 2 and 3. We have provided a clearer explanation on lines 135-146. |
3. | Lines 372-374 - Please amend. Your results, as presented, suggest children can accurately report SOME health and lifestyle behaviours, not a wide range. | This has now been amended to read, “children can accurately report on some health and lifestyle behaviours.” (Lines 444-446). |
4. | Lines 416-420 - This is confusing, especially line 419. Did the CHAT consistently show poor validity or not? | Apologies for the lack of clarity. We have clarified the degree to which children could accurately report valid answers using the CHAT compared to the autographer and accelerometer on lines 444-446. |
5. | Lines 429-431 - The paper does not report the distribution of physical activity across the day so how can the authors claim this as novel and invaluable to the study? | Thank you for this insightful comment. Whilst the study does not report the daily distribution of PA due to the wear-time criteria, it does, however, provide promise of using this methodology to ascertain the total daily distribution of children’s activities in future research. We have now clarified this on lines 499-502. |
6. | Lines 512-516 - this contradicts your abstract and conclusions which now state that your current methodology was not able to validate self-reported behaviours.
| Thank you – we agree and have now ensured clarity over the extent to which children can self-report health and lifestyle behaviours. Given the challenges faced with the current methodology, we acknowledge the sample size was small. Regardless, this study demonstrates that using an autographer in combination with an accelerometer is a novel method to categorise, quantify and describe children’s physical activity, which warrants further research. The revisions can be seen on lines 576-580. |
7. | Lines 529-532 - this contradicts what is written at lines 372-374 | Thank you for highlighting this oversight. Further to the responses above, we have provided clarity throughout the abstract, discussion and conclusion regarding the manuscripts findings. Please see lines 26-32, 444-446, 576-580. |
Round 3
Reviewer 2 Report
Thank you for addressing my comments.